# Achieving Cross Modal Generalization with Multimodal Unified Representation

**Yan Xia**[1*] **Hai Huang**[1*] **Jieming Zhu**[3] **Zhou Zhao**[1,2†]

[1]Zhejiang University [2]Shanghai Artificial Intelligence Laboratory [3]Huawei Noah's Ark Lab

xiayan.zju@gmail.com haihuangcode@outlook.com
jiemingzhu@ieee.org zhouzhao@zju.edu.cn

## Abstract

This paper introduces a novel task called Cross Modal Generalization (CMG), which addresses the challenge of learning a unified discrete representation from paired multimodal data during pre-training. Then in downstream tasks, the model can achieve zero-shot generalization ability in other modalities when only one modal is labeled. Existing approaches in multimodal representation learning focus more on coarse-grained alignment or rely on the assumption that information from different modalities is completely aligned, which is impractical in real-world scenarios. To overcome this limitation, we propose **Uni-Code**, which contains two key contributions: the Dual Cross-modal Information Disentangling (DCID) module and the Multi-Modal Exponential Moving Average (MM-EMA). These methods facilitate bidirectional supervision between modalities and align semantically equivalent information in a shared discrete latent space, enabling fine-grained unified representation of multimodal sequences. During pre-training, we investigate various modality combinations, including audio-visual, audio-text, and the tri-modal combination of audio-visual-text. Extensive experiments on various downstream tasks, i.e., cross-modal event classification, localization, cross-modal retrieval, query-based video segmentation, and cross-dataset event localization, demonstrate the effectiveness of our proposed methods. The code is available at https://github.com/haihuangcode/CMG.

## 1 Introduction

Although recent years have witnessed significant achievements in many multimodal areas, e.g., multi-modality question-answering [1, 2, 3], query-based segmentation [4, 5], audio-visual event localization [6, 7], labeling these tasks always consumes extensive human resources. Moreover, the labeling cost for different modalities can vary significantly [8], leading to scenarios where only a subset of modalities is labeled, while others remain scarce [9]. For example, text-based visual segmentation is common while the label data for audio-based visual segmentation is rare. Consequently, models trained on such limited data are restricted to specific scenarios, hindering their broader applicability. Fortunately, unannotated paired multimodal data is readily accessible, such as the abundant image-caption [10] and audio-visual pairs [11] available on the Internet. Hence, in this paper, we develop a new task, **Cross-Modal Generalization (CMG)**, for investigating how to learn unified discrete representations from these unlabelled multi-modal data pairs. Our objective is to transfer the knowledge acquired from labeled modalities to other unseen modalities in downstream tasks, enabling models to generalize effectively.

---

*Equal Contribution
†Corresponding Author

37th Conference on Neural Information Processing Systems (NeurIPS 2023).

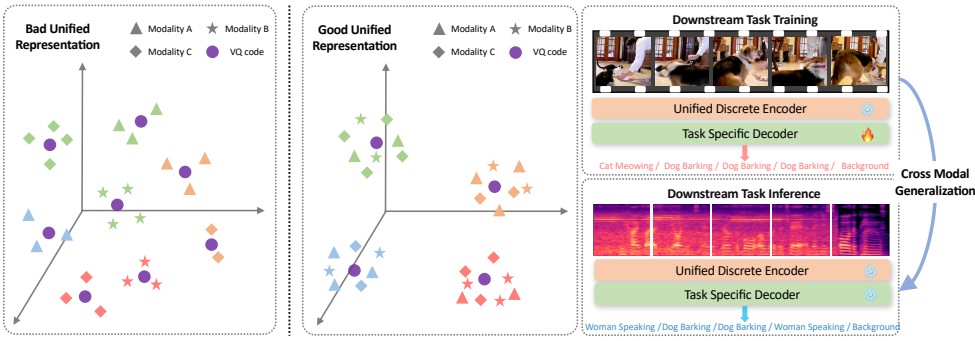

Figure 1: The overview of our proposed CMG task, different colors in the left and middle parts mean different semantics. The left part is the illustration of bad multi-modal unified representation, where features from various modalities sharing the same semantic meaning are mapped into disparate latent codes, while the good unified representation (middle part) is totally different. The right part shows that in downstream tasks, the model will be directly transferred to unseen modalities.

Humans possess a natural ability to associate different modalities with similar semantics, facilitating cross-modal knowledge transfer based on known modalities. Inspired by this, numerous studies have explored the integration of diverse multimodal information into a unified semantic space, which can be categorized into two types: implicit representations [12, 13, 14] and explicit representations [15, 16, 17]. For implicit representations, several methods utilize a modality-agnostic encoder [12, 13] to represent different modalities, or employ contrastive learning [18, 19] to bring different modalities closer in high-dimensional semantic space. In contrast, explicit representations aim to use a unified codebook [17] or prototype [16] to represent different modalities, serving as a bridge to facilitate robust alignment across modalities. Furthermore, the use of discrete space allows for the aggregation of similar input features in a high-dimensional space, enabling complex feature representation with a reduced number of latent codes. However, these studies predominantly compress sequential features within modalities into single vectors before quantization [16, 20] or rely on the assumption that information from different modalities is completely aligned [17, 21]. As a result, these methods are typically limited to simple tasks such as cross-modal retrieval and demonstrate poor performance in more complex scenarios.

As shown in Fig 1, the visual information in unconstrained videos comprises events of *cat meowing* and *dog barking*, while the audio information comprises events of *woman speaking* and *dog barking*. Directly applying previous methods [13, 17, 16, 20] would lead to inaccurate mapping of multimodal information lacking shared semantics. To address this limitation, our paper primarily focuses on the implementation of a fine-grained, unified representation for multimodal sequences. We address this novel problem from two key aspects: **1)** extracting information with identical semantics across different modalities while mitigating the influence of modality-specific details, and **2)** representing these diverse modalities with shared semantics using a unified codebook. Most of the previous works primarily focus on the second aspect while neglecting the importance of the first aspect, which we argue is crucial for achieving cross-modal generalization.

Our work builds upon the advantages of explicit representations and proposes two modules to address the aforementioned challenges. To address the **first aspect**, we propose Dual Cross-modal Information Disentangling (DCID) module, which combines Contrastive Log-ratio Upper Bound (CLUB) [22] with our proposed Cross-modal Contrastive Predictive Coding [23] (Cross-CPC). Concretely, we employ CLUB to optimize the upper bound of mutual information, efficiently distinguishing the modal-agnostic semantic information (conveying primary content or events) from the modal-specific information (additional details unrelated to semantics, such as light and perspective in vision, or timbre and pitch in audio, etc.) across various modalities. However, without effective guidance, it is challenging for the model to identify which aspects of semantic information are useful [24]. Motivated by the complementary guidance nature of multi-modal information [25], we propose a novel Cross-CPC method, to predict the future information in other modalities based on the known sequence information of the current modality, which can effectively maximize the fine-grained mutual information between different modalities. To address the **second aspects**, we propose Multi-Modal Exponential Moving Average (MM-EMA), which employs a teacher-student mechanism to facilitate

cross-modal guidance during pretraining. This mechanism encourages the aggregation of quantized vectors encoded from different modalities with shared semantics into the same latent code space. Otherwise, distinct modalities would aggregate in separate regions of the discrete space, rather than being mapped together [20, 17]. To summarize, our main contributions are threefold:

- We introduce a new task, **CMG**, which enhances the applicability of existing models in the realm of multimodal learning, addressing the challenges arising from modality scarcity and the substantial cost of annotating certain modalities.

- We propose a novel framework named **Uni-Code**, which can effectively extract shared semantic information from paired multi-modal data and project them into a common quantized latent space in a fine-grained level.

- We investigate the unified representation of various modalities, including audio-visual, audio-text, visual-text, and even the challenging tri-modal combination of audio-visual-text. Extensive experiments on various downstream tasks, e.g., multimodal event classification, localization, cross modal retrieval and video segmentation, demonstrate the effectiveness of our proposed methods.

## 2  Related Work

**Implicit Multi-Modal Unified Representation.** The past few years have witnessed remarkable achievements in the implicit multi-modal unified representation, which aims to align diverse modalities within a shared latent space [26, 27, 14] or try to learn a modality-agnostic encoder for extracting information across various modal [28, 29]. These methods investigate the unified representation for various modality combinations, i.e., Speech-Text [26, 30], Video-Audio [31, 32, 27, 33], Vision-Text [18, 14, 29]. To achieve these, different kinds of methods have been proposed. Pedersoli et al. [31] and Sarkar et al. [27] introduce a cross-modal knowledge distillation to transfer knowledge across modalities. CLIP-based methods [18, 14] use contrastive loss to learn image-text consistency from a large paired dataset, and have gained incredible zero-shot ability in various downstream tasks.

**Explicit Multi-Modal Unified Representation.** Recently, some works investigate how to achieve multimodal explicit unified representation by utilizing a universal codebook [15, 17, 20] or prototype [16] to explicitly express multimodal content. Duan et al. [16] apply Optimal Transport to map the feature vectors extracted from different modalities to the prototypes. Zhao et al. [20] use self-cross-reconstruction to enhance the mutual information between different modals. However, these coarse-grained alignment methods [16, 20] are only suitable for simple tasks such as retrieval, and are unable to accomplish fine-grained comprehension in downstream tasks. In recent years, [34, 21] align the text and speech temporally with a unified discrete space. Liu et al. [17] use the same scheme to align short videos and speech/text. However, a key assumption underlying their success is that the modalities they chose have a one-to-one alignment (i.e. text-to-speech), while in more general case, it is hard to guarantee the semantic information in the two paired modalities is completely consistent (i.e. unconstrained video and audio). In this paper, we mainly investigate how to map a paired multimodal sequence into a common discrete semantic space, where the information in the multimodal sequence is unconstrained and may not be perfectly aligned.

**Mutual Information Estimation.** Mutual information (MI) estimation aims to measure the dependence between two random variables. Recent works primarily concentrate on how to combine MI estimation with deep neural networks [35, 22, 23], including MI maximization and minimization. MI maximization aims to learn representations that capture meaningful and useful information about the input data, leading to improved performance in various downstream tasks [36, 37, 38]. In order to maximize mutual information in a sequence, Contrastive Predictive Coding (CPC) [23] employs an autoregressive model and contrastive estimation to capture long-term relations while maintaining local features within a sequence. MI minimization tries to reduce the dependency between two random variables while preserving the relevant information, has been successfully applied in disentangled representation learning [39, 24]. Belghaz et al. [35] propose a Mutual Information Neural Estimator (MINE), which builds a neural network to estimate mutual information based on dual representations of the KL-divergence [40, 41]. Chen et al. [22] introduce Contrastive Log-ratio Upper Bound (CLUB), combining MI estimation with contrastive learning to approximate the MI upper bound. Their work is more suitable for MI minimization.

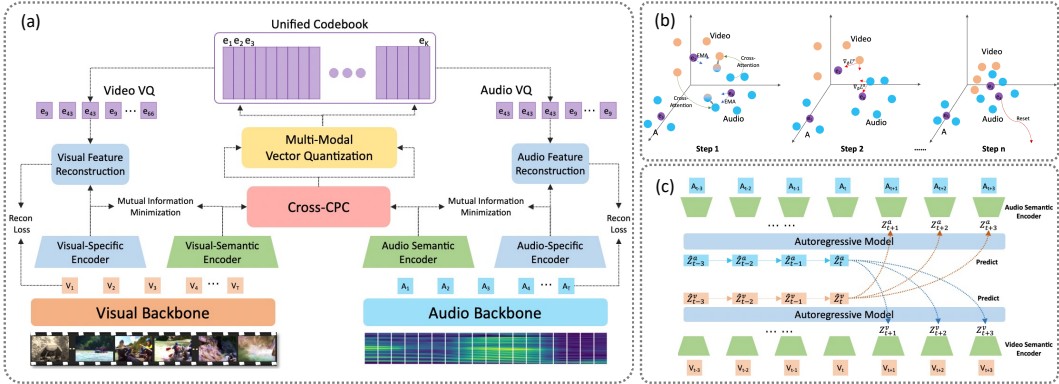

Figure 2: The overview of our proposed Uni-Code framework, we use audio-visual as an example. (a) The main pipeline of our model. (b) The process of Multi-modal Vector Quantization, which contains Multi-modal EMA (MM-EMA) and a new commitment loss, notice that the abundant discrete codes will be reset. (c) The architecture of our proposed Cross-CPC.

# 3 Cross Modal Generalization Task

Given a set of paired multi-modal data $\mathbb{X} = \{(\mathbf{x}_i^A, \mathbf{x}_i^B, \mathbf{x}_i^C...)\}_{i=1}^N$ of size N, where A, B, C, and so on represent different modalities, the Cross Modal Generalization (CMG) task aims to map these various modalities into a unified discrete space during the pre-training phase, enabling discrete latent codes to be shared among different modalities with the same semantic. Subsequently, in downstream tasks, when only one modality (e.g., mode A) has annotated information, the model can transfer knowledge learned from A mode to other modalities (e.g., mode B and C) based on the shared discrete space obtained during pre-training to achieve zero-shot generalization ability.

# 4 Unified Representation Learning

Different from previous works which simply extract information from paired modalities and then direct maps, we argue that the success of unified representation lies in the extraction of modality-agnostic semantic features. Thus in this paper, we achieve this from two perspectives: first, we introduce a DCID module, designed to extract fine-grained semantic information and separate it from the corresponding modality-specific information within each modality. Second, we compress the extracted semantic features into discrete variables through VQ-VAE, ensuring that the compressed discrete variables can still contain the original semantic information through reconstruction loss. For simplicity, we take two modalities as an example to illustrate the process. Fig 2 gives an overview illustration of our network.

## 4.1 Baseline

Given two paired modals, $\{(\mathbf{x}_i^a, \mathbf{x}_i^b)\}_{i=1}^N$, we first utilize two semantic encoders $\Phi^a$ and $\Phi^b$ to extract modal-agnostic features $\mathbf{z}_i^a, \mathbf{z}_i^b \in R^{T \times D}$, and use two modal-specific encoders $\Psi^a$ and $\Psi^b$ to extract remain features $\bar{\mathbf{z}}_i^a, \bar{\mathbf{z}}_i^b$ from modality A and B, respectively:

$$\mathbf{z}_i^a = \Phi^a(\mathbf{x}_i^a) \quad \bar{\mathbf{z}}_i^a = \Psi^a(\mathbf{x}_i^a) \quad \mathbf{z}_i^b = \Phi^b(\mathbf{x}_i^b) \quad \bar{\mathbf{z}}_i^b = \Psi^b(\mathbf{x}_i^b), \tag{1}$$

where T, D represent time and hidden dimension, respectively. The dimension of $\bar{\mathbf{z}}_i^a, \bar{\mathbf{z}}_i^b$ various for different modalities. Then we apply vector quantized operation to map the semantic features $\mathbf{z}_{i,t}^a, \mathbf{z}_{i,t}^b$ to discrete latent codes in fine-grained level, $t \in [0, T]$:

$$\hat{\mathbf{z}}_{i,t}^m = VQ(\Phi^m(\mathbf{x}_{i,t}^m)) = VQ(\mathbf{z}_{i,t}^m) = e_l, \quad \text{where} \quad l = \mathrm{argmin}_j \|\Phi(x) - e_j\|_2, \quad m \in \{a, b\}, \tag{2}$$

the latent codebook $\mathbf{e} \in R^{L \times D}$ is shared across modality A and B, where $L$ is the the size of the discrete latent space. Finally, we combine $\hat{\mathbf{z}}_i^m$ with $\bar{\mathbf{z}}_i^m$ together to reconstruct original features:

$$L = \underbrace{\|\mathbf{x}_i^m - D(\hat{\mathbf{z}}_i^m; \bar{\mathbf{z}}_i^m)\|_2^2}_{\text{reconstruction loss}} + \underbrace{\|\mathrm{sg}[\phi^m(\mathbf{x}_i^m)] - \mathbf{e}\|_2^2}_{\text{VQ loss}} + \underbrace{\beta\|\phi^m(\mathbf{x}_i^m) - \mathrm{sg}[\mathbf{e}]\|_2^2}_{\text{commitment loss}}, \quad m \in \{a, b\}. \tag{3}$$

where $\beta$ is 0.25 for all our experiments, sg is stop gradient. In this work, we use Exponential Moving Average (EMA) to replace VQ loss, since EMA is more robust. The reconstruction loss can guarantee that the compressed latent codes $e_l$ still maintain the semantic information of different modalities. In ideal conditions, $\mathbf{z}_i^a$ and $\mathbf{z}_i^b$ encoded from different modalities with the same semantics should be mapped to the same discrete latent code. However, without effective supervision, the existence of modality gap will result in $\mathbf{z}_i^a$ and $\mathbf{z}_i^b$ converging to separate regions of the codebook [20, 17]. Thus in this paper, we introduce the following modules to alleviate these problems.

## 4.2 Dual Cross-modal Information Disentangling

We introduce our DCID module from two aspects: MI minimization between modal-agnostic semantic features and modal-specific features in each modality (CLUB), and MI maximization between modal-agnostic semantic features across different modalities (Cross-CPC).

**CLUB-based MI Minimization:** Compared to the methods that try to optimize the MI lower bound, such as InfoNCE [23] and MINE [35], CLUB [22] can effectively optimize the MI upper bound, demonstrating superior advantages in information disentanglement [24]. Given two variables $\mathbf{x}$ and $\mathbf{y}$, the objective function of CLUB is defined as:

$$I_{\text{vCLUB}}(\mathbf{x}; \mathbf{y}) := \mathbb{E}_{p(\mathbf{x},\mathbf{y})}[\log q_\theta(\mathbf{y}|\mathbf{x})] - \mathbb{E}_{p(\mathbf{x})}\mathbb{E}_{p(\mathbf{y})}[\log q_\theta(\mathbf{y}|\mathbf{x})], \quad (4)$$

where $q_\theta$ is the variational approximation of ground-truth posterior of $\mathbf{y}$ given $\mathbf{x}$ and can be parameterized by a network $\theta$. We use CLUB to optimize the MI upper bound between the semantic features $\mathbf{z}_i^m$ and modal-specific features $\bar{\mathbf{z}}_i^m$, $m \in \{a, b\}$, we modify $I_{\text{vCLUB}}$ into temporal version:

$$\hat{I}_{\text{vCLUB}} = \frac{1}{N} \sum_{i=1}^{N} \Big[ \frac{1}{T} \sum_{t=1}^{T} \log q_\theta(\bar{\mathbf{z}}_i^m|\mathbf{z}_i^m) - \frac{1}{N}\frac{1}{T} \sum_{j=1}^{N} \sum_{t=1}^{T} \log q_\theta(\bar{\mathbf{z}}_j^m|\mathbf{z}_i^m) \Big], \quad m \in \{a, b\} \quad (5)$$

The approximation network and the main networks are optimized alternatively during pre-training. Finally, we can reduce the correlation between semantic information and modal-specific information in each modality. Nevertheless, merely minimizing mutual information using CLUB presents a challenge for the model to identify relevant semantic features. Given that paired multimodal information can provide mutual guidance and serve as supervisory signals for each other, we devise the Cross-CPC approach to alleviate this issue.

**MI Maximization with Cross-CPC:** Contrastive Predictive Coding (CPC) [23] can maximize the mutual information among the adjacent items within a sequence by predicting the future samples with powerful autoregressive models, has been widely used in self-supervised learning. While for human, we can not only predict subsequent scenarios based on the current modality, but are also capable of associating with potential situations that may occur in other modalities. For instance, people can infer forthcoming auditory information based on a presently viewed video segment or read text, or envision a subsequent scene by perceiving an audio portion. Inspired by this, in this paper, we extend CPC to cross-modal constrictive prediction. Given the semantic features $\mathbf{z}^a, \mathbf{z}^b \in R^{T \times D}$, a prediction of K steps and a random time moment $t \in (0, \text{T-K}]$, we first use two single layers unidirectional LSTM to summarize the information of all $\mathbf{z}_{\leq t}^a, \mathbf{z}_{\leq t}^b$ and can obtain two context representations as $\mathbf{c}_t^m = \text{LSTM}(\mathbf{z}_{\leq t}^m) \in R^D, m \in \{a, b\}$.

For modality A, we first select a set $Z_b$ of N-1 random negative samples and one positive sample $\mathbf{z}_{t+k}^b$ from modality B, then we use $\mathbf{c}_t^a$ to predict k-th future step $\mathbf{z}_{t+k}^b$ in modality B, and the InfoNCE loss for modality A can be optimized as:

$$L_{cpc}^{a2b} = -\frac{1}{K} \sum_{k=1}^{K} log \Big[ \frac{exp(\mathbf{z}_{t+k}^b W_k^a \mathbf{c}_t^a)}{\sum_{\mathbf{z}_j \in Z_b} exp(\mathbf{z}_j^b W_k^a \mathbf{c}_t^a)} \Big]; \quad L_{cpc}^{b2a} = -\frac{1}{K} \sum_{k=1}^{K} log \Big[ \frac{exp(\mathbf{z}_{t+k}^a W_k^b \mathbf{c}_t^b)}{\sum_{\mathbf{z}_j \in Z_a} exp(\mathbf{z}_j^a W_k^b \mathbf{c}_t^b)} \Big], \quad (6)$$

where the $W_k^a$ is the linear projection matrix for different step k. The optimization $L_{cpc}^{b2a}$ for modality B is vice versa. Based on these, features with the same semantics across different modalities can be mutually extracted through fine-grained cross-modal prediction. Despite it is difficult to align paired modalities perfectly in unrestricted scenarios, the model can still predict the possible information in the next step of the other modality at the fine-grained level through contrastive learning, due to the summarization of the historical information using an autoregressive model.

### 4.3 Multi-modal Exponential Moving Average

Previous works [20, 16, 17] struggle to achieve fine-grained cross-modal alignment in unconstrained scenarios, to tackle their limitations, we propose MM-EMA, which can allow different modalities to serve as teacher-student iteratively and update each other during the quantization process.

First, we use cross-attention [42] to extract related information from the opposite modality, taking mode A as an example: $\mathbf{r}_i^b = \text{cross-att}(\mathbf{z}_i^a; \mathbf{z}_i^b; \mathbf{z}_i^b)$, where $\mathbf{z}_i^a$ is query, $\mathbf{z}_i^b$ is key and value. The vector $\mathbf{r}_i^b$ contains the semantic information derived from $\mathbf{z}_i^b$, exhibiting a strong correlation with $\mathbf{z}_i^a$, while also preserving the intrinsic attributes of modality B. Thus, it can serve as an intermediary facilitating the alignment of modality A with modality B during EMA procedure.

Given a code vector $\mathbf{e}_i$, we can obtain $n_i^a$ semantic vectors of modality A $\{\mathbf{z}_{i,j}^a\}_{j=1}^{n_i^a}$ and $n_i^b$ semantic vectors of modality B $\{\mathbf{z}_{i,j}^b\}_{j=1}^{n_i^b}$ that are quantized to $\mathbf{e}_i$:

$$N_i^{(t)} = \gamma N_i^{(t-1)} + (1-\gamma)[n_i^{a(t)} + n_i^{b(t)}] \quad \mathbf{e}_i^{(t)} = \mathbf{o}_i^{(t)}/N_i^{(t)} \tag{7}$$

$$\mathbf{o}_i^{(t)} = \gamma \mathbf{o}_i^{(t-1)} + (1-\gamma)\left[ \sum_{j=1}^{n_i^{a(t)}} \frac{\mathbf{z}_{i,j}^{a(t)} + \mathbf{r}_{i,j}^{b(t)}}{2} + \sum_{j=1}^{n_i^{b(t)}} \frac{\mathbf{z}_{i,j}^{b(t)} + \mathbf{r}_{i,j}^{a(t)}}{2} \right],$$

where t is batch sequence in order, $N_i$ and $o_i$ are accumulated vector count and volume, respectively. Besides, we also modify the commitment loss in Eq 3, which can use the code vector $\mathbf{e}_i^b$ (quantized from $\mathbf{z}_i^b$) as a teacher, to guide the Encoder $\phi^a$ towards not only approximating $\mathbf{e}_i^a$, but also converging to $\mathbf{e}_i^b$ with a certain ratio (50% in our setting, $\beta$ is the same as in Eq 3).

$$L_{commit}^a = \beta\|\phi^a(\mathbf{x}_i^a) - \text{sg}[\mathbf{e}_i^a]\|_2^2 + \frac{\beta}{2}\|\phi^a(\mathbf{x}_i^a) - \text{sg}[\mathbf{e}_i^b]\|_2^2 \tag{8}$$

The modified commitment loss $L_{commit}^b$ is also the same process with modality A. During training, the distance between different modalities in the latent space gradually decreases.

**Reset inactivated code:** In order to alleviate the codebook collapse problem existing in VQ process, which implies that only a subset of codes are extensively utilized while the majority of codes remain inactive, we adopt a code reset strategy: for a given code vector $\mathbf{e}_i$, it will be retained only if it has been selected at least once by both modality A and modality B in $N_{re}$ consecutive batches, whereas the remaining codes will be re-initialized. This strategy can effectively mitigate the issue of some code being perpetually unexploitable due to poor initialization, as well as the problem of generating redundant code during the process of aligning modalities A and B (see appendix for more details).

### 4.4 Training and Downstream Tasks

The full objective of our pre-training framework is the combination of all above objection functions: $L = L_{recon} + L_{commit} + L_{cpc} + L_{cmcm} + L_{MI}$, where $L_{MI}$ is $\hat{I}_{\text{vCLUB}}$, $L_{cmcm}$ is the objective loss proposed in [17], which can also promote the alignment among modalities. All these objectives except $L_{cmcm}$ integrate two modalities. After training, we can obtain a unified discrete latent space.

In this paper, we investigate the unified representation of various modalities, including audio-visual, audio-text, visual-text, and even more, the tri-modal combination of audio-visual-text. Then we can apply the pre-trained multi-modal encoder in various downstream tasks, i.e., cross-modal event classification, event localization, query-based video segmentation and cross both domain & modal event localization. Note that during downstream tasks, the parameters of the pre-trained encoder are frozen.

## 5 Experiments

### 5.1 Datasets and Tasks

**Pre-train:** We use VGGsound-AVEL [44, 45] to pre-train our unified representation, and divide it into several different sizes: 24K, 40K, 81K. Considering that the VGGsound-AVEL dataset only contains audio, video and event label, we design prompts for these labels and modify them into descriptive sentences, see Appendix for more details.

Table 1: Compared with state-of-the-art methods on two downstream tasks. We use precision to indict the performance of the models on AVE tasks, and use accuracy for AVVP tasks.

| Method | VGGsounds-AVEL 24K | | | | VGGsounds-AVEL 40K | | | | VGGsounds-AVEL 81K | | | |
|---|---|---|---|---|---|---|---|---|---|---|---|---|
| | AVE | | AVVP | | AVE | | AVVP | | AVE | | AVVP | |
| | V→A | A→V | V→A | A→V | V→A | A→V | V→A | A→V | V→A | A→V | V→A | A→V |
| Baseline | 4.4 | 5.9 | 7.6 | 8.4 | 5.5 | 5.4 | 6.9 | 8.7 | 7.1 | 9.3 | 5.6 | 7.2 |
| S-Mit[43] | 12.7 | 16.9 | 17.2 | 22.8 | 14.4 | 15.9 | 19.0 | 22.3 | 13.4 | 17.0 | 20.9 | 22.8 |
| MST[13] | 13.3 | 19.0 | 25.7 | 29.1 | 19.5 | 23.1 | 22.7 | 24.5 | 18.6 | 20.5 | 19.1 | 24.8 |
| CODIS[16] | 18.5 | 22.0 | 29.4 | 33.7 | 20.8 | 26.4 | 35.1 | 37.9 | 28.5 | 30.2 | 34.0 | 37.8 |
| TURN[20] | 17.7 | 21.0 | 29.4 | 32.4 | 19.1 | 24.3 | 36.9 | 39.3 | 27.6 | 31.4 | 33.8 | 38.1 |
| CMCM[17] | 28.9 | 35.9 | 42.6 | 50.4 | 32.7 | 36.8 | 41.9 | 45.1 | 31.1 | 34.0 | 39.3 | 44.8 |
| DCID+S-Mit | 28.1 | 32.3 | 45.9 | 49.2 | 32.2 | 34.0 | 47.8 | 53.0 | 34.8 | 37.6 | 51.9 | 53.5 |
| DCID+MST | 31.2 | 35.0 | 50.7 | 52.1 | 34.9 | 37.8 | 54.4 | 59.1 | 33.5 | 35.4 | 57.1 | 59.2 |
| DCID+TURN | 29.4 | 35.3 | 53.4 | 56.0 | 29.7 | 36.9 | 55.2 | 58.2 | 31.9 | 36.8 | 56.2 | 60.9 |
| DCID+CODIS | 33.4 | 36.0 | 53.8 | 60.2 | 36.7 | 41.0 | 52.6 | 62.0 | 35.9 | 40.1 | 54.3 | 59.0 |
| DCID+CMCM | 34.1 | 38.8 | 57.6 | 60.8 | 36.4 | 42.9 | 58.7 | 62.8 | 38.8 | 41.4 | 57.5 | 60.5 |
| **Uni-Code** | **44.0** | **49.7** | **61.9** | **65.7** | **47.7** | **52.3** | **64.0** | **65.6** | **41.2** | **45.6** | **60.5** | **61.7** |

Table 2: Ablation studies of audio-visual pre-training on AVE and AVVP tasks.

| CLUB | Cross-CPC | MM-EMA | Reset code | $L_{cmcm}$ | VGGsounds-AVEL 24K | | | | VGGsounds-AVEL 40K | | | |
|---|---|---|---|---|---|---|---|---|---|---|---|---|
| | | | | | AVE | | AVVP | | AVE | | AVVP | |
| | | | | | V→A | A→V | V→A | A→V | V→A | A→V | V→A | A→V |
| - | ✓ | ✓ | ✓ | ✓ | 34.9 | 35.1 | 50.6 | 54.0 | 37.2 | 40.3 | 52.9 | 59.5 |
| ✓ | - | ✓ | ✓ | ✓ | 4.6 | 5.8 | 10.9 | 24.6 | 5.2 | 7.1 | 12.3 | 24.1 |
| - | - | ✓ | ✓ | ✓ | 29.8 | 34.6 | 30.4 | 32.5 | 35.2 | 36.9 | 32.3 | 34.0 |
| ✓ | ✓ | - | ✓ | ✓ | 34.1 | 38.8 | 57.6 | 60.8 | 36.4 | 42.9 | 58.7 | 62.8 |
| ✓ | ✓ | ✓ | - | ✓ | 37.8 | 41.2 | 59.1 | 61.5 | 38.9 | 40.3 | 55.4 | 62.1 |
| ✓ | ✓ | ✓ | ✓ | - | 39.7 | 42.6 | 58.2 | 62.1 | 41.3 | 46.0 | 58.7 | 62.8 |
| ✓ | ✓ | - | - | ✓ | 28.2 | 30.9 | 41.4 | 49.2 | 31.5 | 33.8 | 46.0 | 48.3 |
| ✓ | ✓ | ✓ | ✓ | ✓ | **44.0** | **49.7** | **61.9** | **65.7** | **47.7** | **52.3** | **64.0** | **65.6** |

**Downstream:** we evaluate the pre-trained models on several downstream tasks using different datasets: **Cross-modal event classification** (AVE [46]): In AVE, each video contains a primary event that is present in both audio and visual information. Therefore, we can train an event classifier using a single modality (e.g., video) and directly evaluate the classifier's performance on another modality (e.g., audio). **Cross-modal event localization** (AVVP [47]): In AVVP, a portion of the data is annotated with fine-grained event labels for both audio and visual modalities. Similar to AVE, we can perform event localization on one modality and then directly transfer the model to the other modality for testing. **Cross both modal and dataset localization/classification**: In order to prove that our model can also achieve cross-modal transfer on different downstream datasets, we use classification tasks for one modality in AVE to train, and directly test fine-grained event localization ability of the model on another modality in AVVP (from AVE to AVVP). Also we test the cross-modal classification tasks between the visual modality in part of UCF-101 (16 classes) and audio modality in part of VGGSound-AVEL (16 classes). **Cross-modal video segmentation** (AVSBench-S4 [48]): In AVS, each piece of AVS data contains a piece of audio, a video clip of the object that emits the sound, a text label corresponding to the object, and a manually labeled mask. We expand each text label into a descriptive sentence. We use one modality (e.g., audio) to train the query-based video segmentation, and directly test the segmentation ability in another modality (e.g., text). **Cross-modal retrieval** (VGGSound-AVEL) We select retrieval task to demonstrate that our method can also be applied to visual-text generalization as well. We use audio as an inter-medium to measure the generalization ability of our model across these two modalities and implement an X-to-audio retrieval task. To be detailed, in the first stage, we train visual-text unified representation learning using VGGSound24K dataset, and then in the second stage, during downstream training, we let the model learn text(video)-audio retrieval, finally during inference, we directly test the generalization ability of the model on video(text)-audio retrieval.

## 5.2 Baselines and Implementation Details

We use the method depicted in Section 4.1 as our baseline. We also compare our methods with several state-of-the-art unified representation methods: MST [13], CODIS [16], S-MiT [43], CMCM [17],

Table 3: Ablation studies of audio-visual-text pre-training on three downstream tasks.

| CLUB | Cross-CPC | MM-EMA | Reset code | $L_{cmcm}$ | VGGsounds-AVEL 40K | | | | | | | |
| | | | | | AVE | | AVVP | | AVE→AVVP | | UCF(v)↔VGG(a) | |
| | | | | | V→A | A→V | V→A | A→V | V→A | A→V | V→A | A→V |
| - | ✓ | ✓ | ✓ | ✓ | 50.2 | 51.8 | 62.4 | 66.2 | 50.1 | 51.2 | 9.87 | 9.59 |
| ✓ | - | ✓ | ✓ | ✓ | 43.8 | 49.2 | 59.3 | 61.1 | 45.5 | 50.6 | 60.6 | 54.6 |
| ✓ | ✓ | - | ✓ | ✓ | 52.9 | 49.9 | 62.0 | 67.3 | 48.1 | 46.8 | 66.5 | 60.5 |
| ✓ | ✓ | ✓ | - | - | 33.0 | 35.5 | 56.7 | 61.2 | 7.4 | 12.6 | 43.3 | 35.2 |
| ✓ | ✓ | ✓ | - | ✓ | 50.8 | 47.8 | 56.4 | 61.1 | 47.9 | 50.4 | 60.0 | 49.8 |
| ✓ | ✓ | ✓ | ✓ | - | 52.4 | 54.5 | 57.5 | **72.9** | 50.5 | 48.7 | **69.9** | 59.7 |
| ✓ | ✓ | ✓ | ✓ | ✓ | **54.1** | **55.0** | **63.4** | 71.0 | **53.0** | **52.4** | 67.1 | **60.6** |
| Evaluation results of the labeled modality | | | | | 64.8 | 65.8 | 71.0 | 72.9 | - | - | 80.0 | 85.4 |

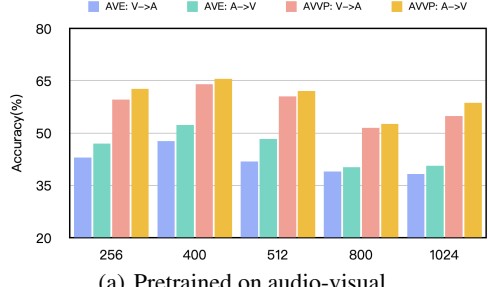

(a) Pretrained on audio-visual.

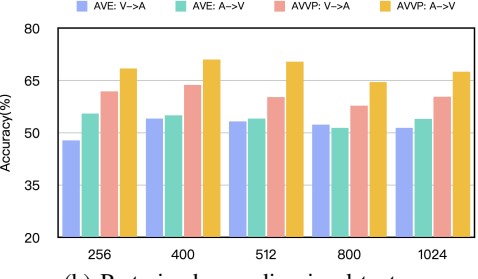

(b) Pretrained on audio-visual-text.

Figure 3: The effect of different codebook sizes on two pre-training tasks.

TURN [20]. We implement these methods on our tasks and test their performance on two downstream tasks. We use the perception to evaluate the performance on AVE, VGGSound-AVEL and UCF101 datasets, and use accuracy (%) to evaluate the performance on AVVP datasets, and use F1-score (%) to evaluate the performance for AVE to AVVP generalization task, and use mIoU and F-score (as same as AVS[48]) for AVS-S4 dataset. The implementation details are provided in Appendix.

## 5.3 Compared with State-of-the-art Methods

We mainly compare our model with these leading methods on two downstream tasks, cross-modal event classification and localization, all these models are pre-trained using audio-visual modalities on three different sizes of datasets. As we can see in Table 1, our methods can outperform all these previous works on three settings. The baseline results indicate that without any constraints, it is challenging for the model to learn the alignment relationship between different modalities during pre-training, leading to poor performance in downstream tasks. Moreover, the experiment results also demonstrate that merely employing these state-of-the-art methods in such unconstrained audio-visual alignment has considerable limitations. However, when incorporated with our proposed DCID module, these models exhibit substantial improvements in downstream tasks. This further suggests that our proposed DCID can effectively decouple the semantic information of different modalities.

In addition, we can observe from Table 1 that when the scale of pre-training data increases from 24K to 40K, there is a significant improvement in the performance of the model on downstream tasks. However, when the scale reaches 81K, the performance of the model does not improve much and even declines. This also indicates that excessive pre-training data may not yield desirable results, only an appropriate amount of data can enable the model to learn the correct alignment relationships.

## 5.4 Ablation Studies

**Main Components** We use the audio-visual unified representation as pre-training task and test the performance on the downstream AVE and AVVP tasks by removing each main component of our model, the results are depicted in Table 2. We can observe that the performances all decline after removing these components. In the DCID module, the impact of removing Cross-CPC is far greater than that of removing CLUB, which indicates that without the fine-grained cross-modal alignment constraints provided by Cross-CPC, it is difficult for CLUB alone to extract effective semantic

information from the original modal features. When MM-EMA is removed, the model performance also declines significantly, demonstrating that MM-EMA can effectively utilize the Cross-Attention mechanism between different modalities to gradually map the relevant information together, helping to achieve a unified representation. The experimental results of removing the "reset code" indicate that this module can effectively remove the redundancy latent codes caused by MM-EMA.

**The Effect of Codebook Size** Different codebook sizes may affect the effectiveness of modality aggregation, thereby influencing the performance of downstream cross-modal generalization tasks. Here we investigate how the size of codebook influences pre-training performance. From Fig 3, we can see that the appropriate codebook size can achieve the best pre-training performance, while too large size (which may cause signals from different modalities to aggregate at different positions in the codebook) and too small size (where semantically distinct signals are erroneously mapped together due to limited capacity) will both degrade the performance of pre-trained models.

**Unified Representation of Three Modalities** Compared with previous works that can only align two modalities together, our method can disentangle and align the semantic information of three modalities, as shown in Table 3. From the results we can see that the removal of these modules will lead to a decline in the performance of the pre-trained model on downstream tasks, similar to the results in Table 2. However, compared to the unified representation of the two modalities, the removal of these modules leads to a smaller decline in the model performance. We think the reason is that the introduction of the third modality can facilitate the other two modalities approaching each other, and the performance of the model pre-trained on three modalities is significantly higher than that of the model pre-trained on two, which also illustrate this point.

**Results on AVS-S4** As we can see in Table 4, our pre-trained model can effectively transfer the segmentation ability across modals. The visualization of segmentation results in Fig 5, our model can accurately identify vocal regions in video frames for unknown modalities. Furthermore, the performance of our method is closer to the evaluation results of the AVS model on source modality. More visualization results and analysis can be found in Appendix.

**Results on Cross Modal Retrieval Tasks** We also test the performance of X-to-audio retrieval task under cross modal generalization setting, where X can be visual or text, the results are as shown in Table 5. Compared with baseline model, our methods can greatly advance the audio retrieval ability in both two generalization directions.

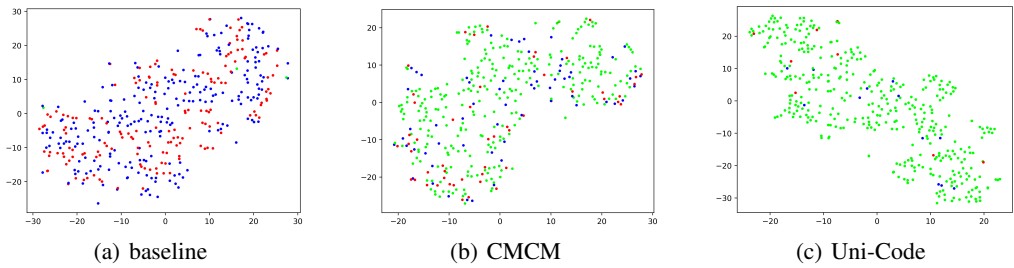

|         (a) baseline          |          (b) CMCM          |         (c) Uni-Code          |

Figure 4: Visualization of the discrete codes. The blue dot means that the number of visual modal quantization accounts for more than 80% of the total number of audio and video quantification, the red dot means that the audio station accounts for more than 80%, and the green dot means that audio and video each account for more than 20%. As we can seen in (b), the discrete codes obtained by CMCM still contain a little single modality mapped codes, while ours (c) are almost all green codes which can be mapped by both two modalities.

## 5.5 Qualitative Analyses

To further illustrate the effectiveness of our model during pre-training phase, we take audio and video as an example and visualize the latent discrete codes obtained from two modalities. As we can see in Fig 4(a), when there is no additional supervisory signal, different modalities are difficult to map to the same discrete code, which also proves why the baseline performs poorly in downstream tasks. The CMCM [17] method can effectively alleviate this problem, but there are still many codes that can only be mapped by single modality. However, our model enables information from different modalities but with the same semantics to be mapped into the same codes. (as shown in Fig 4(c)).

Table 4: Performance on AVS-S4 datasets (pretrained on audio-visual-text modalities).

| Methods | A2T | | T2A | |
|---|---|---|---|---|
| | mIoU | F-score | mIoU | F-score |
| Baseline | 69.8 | 81.4 | 69.9 | 81.3 |
| Our full model | **78.0** | **87.1** | **77.7** | **86.7** |
| SST [49] (A2A) | 60.3 | 80.1 | - | - |
| AVS [48] (A2A) | 78.7 | 87.9 | - | - |

Table 5: Performance of audio retrieval tasks under cross modal generalization directions.

| Methods | V2T | | T2V | |
|---|---|---|---|---|
| | R@5 | R@10 | R@5 | R@10 |
| Baseline | 0.47 | 1.03 | 0.62 | 0.85 |
| Our full model | **10.3** | **21.9** | **8.47** | **16.7** |

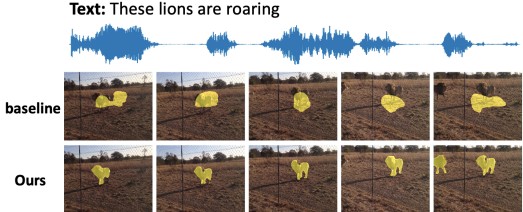

Figure 5: Visualization results of A2T (left) and T2A (right) of our model on AVS-S4 dataset. We compare our method with the baseline model.

## 6   Conclusion and Discussion

In this paper, we propose a new Cross Modal Generalization task, and bring up two innotative modules, DCID and MM-EMA, which can achieve multi-modal unified representation. We achieve unified representation of different multi-modal combinations, including audio-visual, audio-text and audio-visual-text. Also, we conduct extensive experiments on various downstream tasks. The extensive experiments demonstrate the effectiveness of our proposed methods. Furthermore, our model can help existing models to extend into other modalities, achieving more extensive applications. **Limitations and Boarder Impact:** In our work we only focus on the unified representation of three modalities. However, our method can inspire future works to explore the combination of more modalities. Our work do not contain potential negative social impact.

## 7   Acknowledgments

This work is supported by National Key R&D Program of China under Grant No.2022ZD0162000, National Natural Science Foundation of China under Grant No. 62222211 and No.61836002. We also gratefully acknowledge the support of MindSpore (https://www.mindspore.cn), which is a new deep learning computing framework.

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
