# OpenReview forum: "Achieving Cross Modal Generalization with Multimodal Unified Representation"
_NeurIPS.cc/2023/Conference — NeurIPS 2023 poster_

### Official Review · Reviewer_w8Lb · 2023-07-03

**Soundness:** 3 good
**Presentation:** 3 good
**Contribution:** 3 good
**Rating:** 5
**Confidence:** 4

**Summary:**

This paper is focused on a novel and promising setting, which is the zero-shot generalization ability in other modalities that lacks annotations. It is meaningful in real-world scenarios even though it still requires paired multimodal pretraining. In addition, I agree with the authors that the ability of the network for unseen modalities is significant to the community, and the paradigm without fine-tuning encoders in the downstream tasks is also worth more attention.

**Strengths:**

- Useful insights and meaningful technical direction
- Promising experimental results and detailed evaluation
- Detailed and sufficient analyses

**Weaknesses:**

- The organization of the content. The manuscript lacks more promising proposals in the introduction. Obviously, the paragraphs in L58-75 can be greatly reduced.
- The visualization in Figure 4 contains very few instances. It's suggested to present a more comprehensive comparison.
- This paper is more focused on audio-visual understanding and introduces text to these tasks. The core contribution of generalizing to unseen modalities is not sufficiently evaluated.
- The ability of the network for zero-shot learning on seen modality is not evaluated. With strong pretraining based on paired data, I think that it has a strong ability for zero-shot learning. I strongly encourage the authors to add two significant experiments: i. Zero-shot classification on ImageNet ii. Zero-shot classification on AudioSet. It will significantly improve the paper's quality and impact, and I will consider improving my rating.

**Questions:**

- According to Figure 3, it seems that text plays an important role in pretraining. It makes the contributions weaker, for which the improvements of this method can be attributed to the paired data of text, audio, and image. In other means, the dependency on paired data still limits the performance although the authors aim to reinforce the ability of unseen modality generalization.

**Limitations:**

The limitation of this paper is mainly about current applications on audio-visual tasks.

---

> ### Author Rebuttal · Authors · 2023-08-08
>
> Dear Reviewer w8Lb:
>
> Thank you very much for taking time to read our paper and giving such insightful comments. Please see the following for your point-by-point response.
>
> ---
>
> **Weakness 1. The organization of the introduction**
>
> We greatly appreciate your useful comments to make our paper better. We have reorganize the description of the introduction, especially the content you mentioned in Line 58-75.
>
> ---
>
> **Weakness 2. The visualization in Figure 4 contains very few instances.**
>
> Sorry about the confusion, actually, we provide more visualization results of our model and baseline model in different training epochs, as shown in Figure 2 in appendix material. The results can more intuitively illustrate the process of our model gradually aligning different modalities in the latent space.
>
> ---
>
> **Weakness 3. This paper is more focused on audio-visual understanding and introduces text to these tasks.**
>
> Sorry about the confusion, in this paper, we do not focus on audio-visual understanding task, instead, we use various audio-visual tasks to demonstrate that our pre-trained model can directly generalize from seen modality to unseen modality. For example, in **cross-modal event localization** task, after obtaining pre-trained unified representation model, we use it as Encoder to train the event classification task on seen modality (e.g., audio), then during inference, we directly test the performance of event classification ability on unseen modality (e.g., video). Also we conduct downstream experiments on AVS-S4 dataset to demonstrate that our model can achieve zero-shot cross-modal (text2audio and audio2text) segmentation ability.
> We think such these cross-modal generalization tasks can demonstrate that our method can successfully achieve multi-modal unified representation.
>
> ---
>
> **Weakness 4. The ability of the network for zero-shot learning on seen modality is not evaluated.**
>
> In this paper, the purpose of our pre-training is to align different modalities closer in the hidden embedding space, by using a set of intermediate discretized codes. Thus, after pre-training, different modalities but share the same semantics will aggregate to the same discrete latent codes. However, these discrete variables do not contain any knowledge of downstream tasks. To achieve zero-shot cross-modality knowledge transfer, the model has to train these discrete vectors using seen modality during downstream training. In conclusion, our model can not achieve zero-shot learning on seen modality.
>
> However, our pre-train model can not only align different modalities, but also can converge features with similar meaning in the same modality into the same discrete variable. Thus, our model can achieve promising results in few-shot ability on seen modality in downstream tasks.
>
> We conduct a simple few-shot experiments on UCF-101 dataset (video) and part of AudioSet (audio), the model is pre-trained on VGGSound-AVEL 40K.
>
> **Video (UCF-101 dataset):**
>
> | Model         |1-shot |2-shot |4-shot |8-shot |16-shot|32-shot|
> |---------------|-------|-------|-------|-------|-------|-------|
> | baseline      | 1.20  | 1.18  | 1.23  | 1.21  | 1.23  | 1.22  |
> | w/o CrossCPC  | 18.6  | 23.4  | 29.3  | 33.3  | 36.2  | 37.7  |
> | Full model    | 26.1  | 33.2  | 39.0  | 43.2  | 45.9  | 47.3  |
>
> **Audio (AudioSet dataset):**
>
> | Model         |1-shot |2-shot |4-shot |8-shot |16-shot|32-shot|
> |---------------|-------|-------|-------|-------|-------|-------|
> | baseline      | 1.17  | 1.23  | 1.21  | 1.23  | 1.24  | 1.17  |
> | w/o CrossCPC  | 23.6  | 31.2  | 38.0  | 42.8  | 46.5  | 49.2  |
> | Full model    | 32.4  | 44.5  | 51.8  | 56.9  | 58.0  | 60.6  |
>
>
> From the results we can see that our method can outperform the baseline by a large margin under all setting, which can further illustrate that our model can aggregate the similar semantic features, whether they come from the same modality or different modalities. We thank the reviewer again for pointing this out, which can further improve the quality of our work. We hope the additional experiments will address your concern. We will add these few-shot experiments and analysis in the revised paper.
>
> ---
>
> **Question 1. According to Figure 3, it seems that text plays an important role in pretraining**
>
> Thank you very much for your careful observation, let we explain your question in detail. As you can see in Figure 3, the introduction of text will indeed improve the unified representation of audio-visual. This is because the introduction of a third-party modality will serve as a bridge to help the alignment of the other two modalities, which has also been mentioned in other papers, such as UniVAL [1]. When introducing the third mode, our proposed Cross-CPC and MM-EMA will let any two modes interact with each other to shorten the distance, which can facilitate better alignment among these modalities. The performance gain brought by the text does not mean that our contribution is weakened.
>
> [1] UnIVAL: Unified Model for Image, Video, Audio and Language Tasks
>
> ---
>
> **Question 2. In other means, the dependency on paired data still limits the performance although the authors aim to reinforce the ability of unseen modality generalization.**
>
> Sorry about the confusion. In addition, we conduct a series of experiments on unpaired downstream datasets: transferring the event localization ability of the model from **seen modality in AVE dataset** to **unseen modality in AVVP dataset** (Line 242-245), as shown in the right part in Table 3. The results prove that even though the seen and unseen modalities come from different sources, our method can still guarantee a strong zero-shot cross-modal generalization ability with unsupervised pretraining. We will add more analysis about these in our paper.
>
> ---
>
> We thank the reviewer again for your positive feedback. If you have any further questions or comments, please let us know, we are glad to respond.

---

> > ### Comment · Reviewer_w8Lb · 2023-08-15
> >
> > I thank the authors for responding. After carefully reading these responses, I think that this paper can satisfy the standard of publications in NeurIPS. I will keep my rating and look forward to the presentation of this paper.

---

> > > ### Author Response · Authors · 2023-08-19
> > >
> > > Thank you very much! We are glad to know that our response addresses your concerns. We will revise our paper based on your insightful suggestions.

---

### Official Review · Reviewer_bPma · 2023-07-04

**Soundness:** 3 good
**Presentation:** 3 good
**Contribution:** 3 good
**Rating:** 5
**Confidence:** 2

**Summary:**

This paper proposes to learn a unified discrete representation from paired multimodal data during pre-training. During the downstream task, it can achieve zero-shot generalization ability in other modalities when only one modal is labeled. Specifically, it develops a Dual Cross-modal Information Disentangling (DCID) module and a Multi-Modal Exponential Moving Average (MM-EMA) to achieve the goal. Experiments are conducted on four tasks.

**Strengths:**

1. This paper is well-written and easy to follow.
2. The motivation is reasonable.
3. The proposed method is technically sound.
4. Experiments are sufficient.

**Weaknesses:**

1. The compared methods are out-of-date. The authors should provide more latest works for comparison.
2. Missing some related work of unified representation learning.

**Questions:**

see above

**Limitations:**

See above

---

> ### Author Rebuttal · Authors · 2023-08-08
>
> Dear Reviewer bPma:
>
> Thank you very much for your acknowledge of our paper. We are glad to answer your questions point by point.
>
>
> **Weakness 1. The compared methods are out-of-date. The authors should provide more latest works for comparison**
>
> Multi-modal unified representation is a challenging task which has not been studied much. The methods we compared are all the latest studies in explicit unified representation, CODIS (CVPR 22'), CMCM (ACL 22'), TURN (NeurIPS 22'). To achieve cross-modal generalization, discrete variables that can explicitly aggregate different modalities together are necessary. Thus in the experiments, we only compare our model with these latest explicit unified representation methods. Meanwhile, we will add more discussion of the latest implicit representation methods in our paper.
>
>
> **Weakness 2. Missing some related work of unified representation learning**
>
> Thank you very much for pointing this out. We will discuss some recent excellent implicit unified representation models in the related works. For example, as the Reviewer WtGp suggested, the MAE-based (MAViL, CAV-MAE) and CLIP-based (AVE-CLIP, AV-CLIP) multi-modal models, ImageBind, Zorro and XKD. All these works can align different modalities closer in latent embedding space. We will add the discussions of these works in introduction and relate work parts.
>
> We thank the reviewer again for your positive feedback. If you have any further questions or comments, please let us know, we are glad to respond.

---

> > ### Comment · Reviewer_bPma · 2023-08-11
> > **Thanks for authors' response**
> >
> > I have carefully read the authors' responses and other reviewers' comments. I think the authors address my concerns. I will keep my rating.

---

> > > ### Author Response · Authors · 2023-08-19
> > >
> > > Thank you very much! We are glad to know that our response addresses your concerns.

---

### Official Review · Reviewer_y1JE · 2023-07-05

**Soundness:** 3 good
**Presentation:** 3 good
**Contribution:** 3 good
**Rating:** 5
**Confidence:** 4

**Summary:**

The paper proposes a model to learn a unified discrete representation from paired multimodal data during pre-training. Then in downstream tasks, the model can achieve zero-shot generalization ability in other modalities when only one modality is labeled. The two key contributions are the Dual Cross-modal Information Disentangling (DCID) module and the Multi-Modal Exponential Moving Average (MM-EMA). These methods facilitate bidirectional  supervision between modalities and align semantically equivalent information in a shared discrete latent space, enabling fine-grained unified representation of multimodal sequences. Extensive experiments on various downstream tasks show the effectiveness of their method.

**Strengths:**

1. The authors performed extensive experiments to demonstrate the effectiveness of their method.
2. Their method can be applied to more than two modalities and can achieve zero-shot generalization ability in other modalities when only one modality is labeled in downstream tasks.


**Weaknesses:**

1. There are still some parts needs to be clarified, see Questions part.
2. It seems the new commitment loss (eq 8) is not in the ablation study. How does it compare with the original commitment loss in eq(3)?
3. It would be better to include same modality upper bound in downstream tasks

**Questions:**

1. Sg is not defined in eq3
2. How to select the negative set Zb is not clear. Is the size N-1? Then where does the randomness come from? In time steps?
3. The text prompts (Sec B.3) seems to be manually designed and are very specific to the event and datasets. Are there ways to easily generate prompts across different events in different datasets?
4. It would be better to also show some qualitative video segmentation examples of the baseline (compared) methods.

**Limitations:**

The authors have addressed the limitations of their work

---

> ### Author Rebuttal · Authors · 2023-08-08
>
> Dear Reviewer y1JE:
>
> Thank you so much for taking time to read our paper and providing valuable comments. We are glad to respond your questions point-by-point.
>
>
> **Weakness. The comparison with original commitment loss**
>
> Thank you for pointing out our missing ablations, we conduct a series of experiments with the model pretrained on VGGSound 40K to illustrate these:
>
> Cross-modal event classification (AVE dataset):
>
> | Model         | V->A  | A->V  |
> |---------------|-------|-------|
> | Full Model    | 47.7  | 52.3  |
> | Eq(8)->Eq(3)  | 45.1  | 48.2  |
>
> Cross-modal event localization (AVVP dataset):
>
> | Model         | V->A  | A->V  |
> |---------------|-------|-------|
> | Full Model    | 64.0  | 65.6  |
> | Eq(8)->Eq(3)  | 59.6  | 62.3  |
>
> The results show that compared with original Eq (3), our new proposed commitment loss can achieve better multi-modal unified representation. We hope this additional experiment could strengthen our contribution.
>
>
>
> **Question 1. Sg is not defined in eq3**
>
> Sorry about this mistake. Sg in Eq (3) and (8) is the abbreviation of **Stop Gradient** that blocks gradients from flowing into its argument. We will revise this in our paper.
>
> **Question 2. How to select the negative set Zb is not clear**
>
> Yes, as we mentioned in Line 186, the size of negative samples is **N-1**, and these negative samples are randomly selected from other sequences within the same batch. Thank you for noticing this detail, we will add more clarification in our paper.
>
>
> **Question 3. About the design of the text prompts**
>
> Yes, it is a good question. In our paper, for simplified, we manually design several different prompt templates for each event category in VGGSound-AVEL and AVS-S4 datasets. Under your suggestion, we try to use ChatGPT to automatically generate a variety of corresponding prompts according to the characteristics of each event, and the quality is satisfactory.
>
> **Question 4. It would be better to also show some qualitative video segmentation examples of the baseline methods.**
>
> Thank you for pointing this out, we have added the qualitative video segmentation results of the baseline model. Please refer to the global PDF we just uploaded. We will also update the Figure 5 in our paper.
>
>
> We thank the reviewer again for your positive feedback. If you have any further questions or comments, please do not hesitate to ask, we are glad to respond.

---

> > ### Comment · Reviewer_y1JE · 2023-08-15
> >
> > Thanks for the authors' response. I have read the rebuttal and other reviewers' comments. I think they address my concerns. I will keep my rating.

---

> > > ### Author Response · Authors · 2023-08-19
> > >
> > > Thank you very much! We are glad to know that our response addresses your concerns. We will revise our paper based on your insightful suggestions.

---

### Official Review · Reviewer_WtGp · 2023-07-06

**Soundness:** 3 good
**Presentation:** 4 excellent
**Contribution:** 2 fair
**Rating:** 6
**Confidence:** 4

**Summary:**

In this paper the authors propose a novel task called Cross Modal Generalization (CMG), where they aim to use unlabelled internet scale paired multimodal data during pre training and then use it for zero shot generalization to other modalities in downstream tasks. The authors claim that disentagling modality specific features is crucial for allowing shared representations to generalize across modalities for which they propose a Dual Cross-modal Information Disentangling (DCID) module which incorporates 2 different aspects ie MI minimization between modal-agnostic semantic features and modal-specific features in each modality (CLUB) and MI maximization between modal agnostic semantic features across different modalities (Cross-CPC). They further propose Multi-modal Exponential Moving Average (MM-EMA)  to achieve fine-grained cross-modal alignment in unconstrained scenarios.
The authors demonstrate the effectiveness of their approach on several downstream tasks and show significant improvements over the prior state of the art on standard benchmarks. They also present an elaborate ablation study to demonstrate the contributions of each of the components of their proposed approach and show how their proposed modules improve upon an otherwise weak baseline. They present their results on AVE, AVVP and AVS-S4 datasets and show the effectiveness of their approach on Audio-video, audio-text and audio-video-text modalities.

**Strengths:**

- The paper is very well written and addresses several major challenges in the field of learning unified multimodal representations. It's claim of the necessity to disentangle modality specific features is well founded in literature and the method proposed to achieve it solid and sound.
- Thorough ablation studies are performed to highlight the use of each of the proposed method components
- Presentation of the different components and the overall setup is easy to follow and understand
- I like the formulation of the Dual Cross-modal Information Disentangling module. It takes cues from MI maximization theory of disentaglement and is able to formulate it in a simple and succint manner while being able to carefully optimize and train using a widely diverse training objective.

**Weaknesses:**

- The paper doesnt include comparisons to a vast body of literature of Masked Autoencoders which have recently gotten very popular for learning joint vector embeddings. For example: Contrastive Audio-Visual Masked Autoencoder by Gong et al, MAViL: Masked Audio-Video Learners by Huang et al, Masked Autoencoders Are Scalable Vision Learners by He et al, Masked Autoencoders that Listen by Huang et al etc. Similarly the CLIP style approaches and their extensions to further modalities: AVE-CLIP, AV-CLIP, Audio-CLIP etc have not been discussed and compared against. Both these lines of methods have proven to be very effective in literature and should make for good comparison cases here (or discussed as to why they arent relevant comparisons for this work). Another recent work which expands to even more modalities is ImageBind: One Embedding Space To Bind Them All by Girdhar et al and should be compared against
- The suite of downstream tasks to be evaluated against need to be more diverse to truly evaluate if the semantic modality agnostic features have truly been captured by the representations.
- The work seems to be able to easily extended to text-visual modality as well for which there are several benchmarks and baselines available to compare against and would have made for a good comparison but that extension was never done. Curious as to why that combination wasnt considered?

**Questions:**

- The work seems to be able to easily extended to text-visual modality as well for which there are several benchmarks and baselines available to compare against and would have made for a good comparison but that extension was never done. Curious as to why that combination wasnt considered?
-


**Limitations:**

- No major negative impacts or limitations

---

> ### Author Rebuttal · Authors · 2023-08-08
>
> Dear Reviewer WtGp:
>
> Thank you very much for your insightful comments and your acknowledgement of our proposed Dual Cross-modal Information Disentangling module. Let us illustrate your questions point by point.
>
> ---
>
> **Weakness 1. For comparison with MAE-based and CLIP-based method**
>
> We much appreciate the reviewer for pointing out these important methods we missed. As you mentioned, the **MAE-based** and **CLIP-based** audio-visual methods can effectively align two modalities sharing the same semantics within a joint embedding space. These implicit unified representation methods can greatly advance the performance of cross-modal retrieval and multi-modal fusion tasks. However, these methods can not explicitly map different modalities together in the latent space, which may not obtain optimal performance in our cross-modal generalization downstream tasks. For example, we implement AVE-CLIP pre-trained methods on VGGSound-AVEL 40k and compare its performance with our methods on cross-modal event classification task (AVE dataset):
>
> | Model               | V->A | A->V |
> |------------------|-------|------|
> | AVE-CLIP         | 24.8  | 27.8  |
> | CMCM             | 32.7  | 36.8  |
> | Our Full Model | 47.7  | 52.3  |
>
> From the results we can observe that such contrastive learning between audio-visual modalities is not sufficient for perfect multi-modal unified representation.
>
> In recent years, explicit unified representation methods such as CODIS (CVPR 22'), TURN (NeurIPS 22'), and CMCM (ACL 22') have been introduced. These methods utilize discrete latent variables as bridges to explicitly merge different modalities into quantized codes, making them more suitable for this new task. Furthermore, TURN (NeurIPS 22') also contains a self-cross-reconstruction mechanism similar to MAE, we run their code on our task, and the results in Table 1 show that our methods can greatly suppress them. We much appreciate the reviewer for pointing this out, we have added these discussions of these methods and ImageBind in our paper.
>
> ---
>
> **Weakness 2. The suite of downstream tasks to be evaluated against need to be more diverse to truly evaluate if the semantic modality agnostic features have truly been captured by the representations.**
>
> Having said that, it is a very constructive suggestion. We totally agree with the reviewer for this opinion. In our paper, we conduct following experiments to demonstrate it:
> 1) We incorporate our DCID module with other state-of-the-art models, and the results in Table 1 show that with the equipment of DCID, all these model can obtain substantial improvements in downstream tasks, which can illustrate that our DCID method can effectively extract modality-agnostic features and facilitate all other compared models.
>
> 2) Furthermore, as shown in Figure 5, the visualization results illustrate that although there are huge domain gaps between text and audio modalities, the model trained based on text (or audio), can still accurately localize the right visual region when the query been replaced as audio (or text) during inference. These experiments show that our pre-train model can **effectively disentangle semantic modality agnostic features** from text and audio modalities.
>
> Thank you very much for your valuable suggestions. We will design more diverse experiments to demonstrate that our model can truly capture the semantic modality agnostic features.
>
> ---
>
> **Weakness 3 & Question. More experiments about text-visual cross modal generalization tasks**
>
> We appreciate the reviewer for such constructive comment. Yes, our work can be extended to text-visual modality. Here we select **retrieval task** to demonstrate that our method can also be applied to **visual-text generalization** as well. We use audio as an inter-medium to measure the generalization ability of our model across these two modalities and implement an X-to-audio retrieval task. To be detailed, in the first stage, we train **visual-text unified representation** learning using VGGSound24K dataset, and then in the second stage, during downstream training, we let the model learn text(video)-audio retrieval, finally during inference, we directly test the generalization ability of the model on video(text)-audio retrieval. To simplify, we choose cross-attention network as our downstream retrieval model.
>  We test the retrieval performance on part (8k) of AudioSet dataset, the results are as follows:
>
>
> v->t (v2a retrieval for training, t2a retrieval for test)
>
> | Model         | R@5   | R@10  |
> |---------------|-------|-------|
> | Baseline      | 0.47  | 1.03  |
> | Full Model    | 10.3  | 21.9  |
>
>
> t->v (t2a retrieval for training, v2a retrieval for test)
>
> | Model         | R@5   | R@10  |
> |---------------|-------|-------|
> | Baseline      | 0.62  | 0.85  |
> | Full Model    | 8.47  | 16.7  |
>
> The results show that compared with baseline model, our model can effectively achieve **zero-shot cross-modal audio retrieval** ability.
> Due to the rebuttal time constraints, here we only compare our full model with baseline model, however, we will further perform visual-text benchmarks on other comparison methods and add these experiments to our paper. We really appreciate your comments, which make our paper more solid.
>
> ---
>
> We thank the reviewer again for raising these insightful suggestions to make our paper better. If you have any further questions or comments, please let us know, we are glad to respond.

---

### Official Review · Reviewer_jARS · 2023-07-06

**Soundness:** 3 good
**Presentation:** 3 good
**Contribution:** 3 good
**Rating:** 7
**Confidence:** 3

**Summary:**

This paper first introduces a new task called Cross Modal Generalization, which aims to learn a unified discrete representation from paired multi-modal data during pre-training, and realize zero-shot generalization in other modalities in downstreams tasks. This paper proposes Dual Cross-modal Information Disentangling (DCID) module and Multi-Modal Exponential Moving Average (MM-EMA) to facilitate bidirectional supervision between modalities and align semantically equivalent information in a shared discrete latent space. Experiments on various downstream tasks validate the effectiveness of the proposed methods.

**Strengths:**

1. Introduce a new task CMG, mapping various modalities into a unified discrete space.
2. Propose DCID and MM-EMA, extracting shared semantic information and project them into a common quantized latent space.
3. Performance significantly outperforms previous methods.

**Weaknesses:**

1. This paper only perform modality transfer on one pair of modalities, A & V.

**Questions:**

1. The subfigure (b) and (c) in Figure 4 do not look much different. May consider a better visualization, such as the color of the points.
2. "sg" in Equation (3) and (8) is not explained.

**Limitations:**

The authors mention limitations that they only focus on the unified representations of three modalities and future works can explore more modalities.

---

> ### Author Rebuttal · Authors · 2023-08-08
>
> Dear Reviewer jARS:
>
> We appreciate your positive feedback and providing very valuable suggestions. Let us respond to your questions point by point.
>
> ---
>
> **For Weaknesses: This paper only perform modality transfer on one pair of modalities, A & V.**
>
> Thanks for asking! Yes, most of our pretraining and downstream experiments are conducted on paired audio-visual dataset. However, to further demonstrate the effectiveness of our methods, we also pretrain our model on **audio-text** and **audio-visual-text** combinations. And the results on **downstream referring video segmentation** tasks (see Table 4, Figure 5 and Line 301-305) illustrate that our methods can also successfully learn a unified representation of audio & text, and transfer the video segmentation ability from seen modality (e.g., text) to unseen modality (e.g., audio) on AVS-S4 dataset.
>
> In addition, we conduct a series of experiments on **unpaired downstream datasets**: transferring the event localization ability of the model from **one modality in AVE dataset** to **unseen modality in AVVP dataset** (Line 242-245), as shown in the right part in Table 3. The results prove that even though the seen and unseen modalities come from different sources, our method can still guarantee a strong zero-shot cross-modal generalization ability with unsupervised pretraining. We will add more analysis about these in our paper.
>
> Furthermore, according to the suggestion of Reviewer WtGp, we also add a simple cross-modal retrieval experiment to demonstrate that our model can also be extended to visual-text. Please see the ''**For Weakness 3 & Questions**'' section of the reply to reviewer WtGp for more experimental details.
>
> We hope these analysis and new experiments could address the reviewer’s concern and strengthen our paper.
>
> ---
>
> **For Question 1: The subfigure (b) and (c) in Figure 4 do not look much different.**
>
> Thank you very much for your constructive suggestions! As you mentioned, although our model can achieve better multi-modal alignment result than CMCM method, the difference between subfigure (b) and (c) is not obvious. We have replaced the color of both two modalities mapped codes **from purple to green**, please note that we have added these visualizations to the newly updated global PDF.
>
> ---
>
> **For Question 2. "sg" in Equation (3) and (8) is not explained.**
>
> Thank you for pointing this out. We apologize for omitting such important information. Sg in Eq (3) and (8) is the abbreviation of **Stop Gradient** that blocks gradients from flowing into its argument. We will revise this in our paper.
>
> ---
>
> We thank the reviewer again for your positive feedback. We would be very grateful if the reviewer could take time to read our responses and let us know your thoughts.

---

### Official Review · Reviewer_kKgn · 2023-07-25

**Soundness:** 4 excellent
**Presentation:** 4 excellent
**Contribution:** 4 excellent
**Rating:** 8
**Confidence:** 4

**Summary:**

The paper proposes a new pretraining task called Cross Modal Generalization (CMG) for learning unified multimodal representations. The goal is to map different modalities (e.g. audio, visual, text) to a shared discrete latent space during pretraining, such that the model can generalize to unseen modalities in downstream tasks when only one modality is labeled. The paper solves two issues: (1) unified semantic features shared cross-modalities. (2) representing these semantic features using a unified codebook. The paper mainly contributes to the first aspect.

**Strengths:**

- New task is brought to the community and the task makes sense to me because in reality, only partial data is labeled and other with them is largely not.
- DCID and MM-EMA are intuitive methods for disentangling and aligning semantic information across modalities. Using mutual information optimization and teacher-student aggregation.
- Experiments and ablation studies are quite solid.

**Weaknesses:**

- I didn't find weaknesses.

**Questions:**

- In the formula (1), are the $\Phi^{a}$ and $\Phi^{b}$ the same in the implementation?

**Limitations:**

- None.

---

> ### Author Rebuttal · Authors · 2023-08-08
>
> Dear Reviewer kKgn:
>
> Thank you very much for your acknowledgement of our methods and giving us positive feedback! We will respond to your question as follows:
>
> **In the formula (1), are the ${\Phi}^{a}$ and ${\Phi}^{b}$ the same in the implementation?**
>
>  Sorry about the misunderstanding. For different modalities, ${\Phi}^{a}$ and ${\Phi}^{b}$ are not always the same in the implementation. For a given visual feature $V \in R^{B\times T \times H \times W \times C}$, we first apply average pooling operation on it and obtain $\bar{V} \in R^{B\times T \times C}$. We follow the Spatial-Channel Attention proposed in CMRAN [1], which use the $\bar{V}$ to calculate attention score with $V$ in spatial and channel level, then we apply self-attention in temporal level and get the final visual semantic results as $Z^{v} \in R^{B\times T \times C}$. For audio and text input feature, we directly use a self-attention layer in temporal level to get the final audio or text semantic features as $R^{B\times T \times C}$. Thank you very much for pointing this out, we will add these implementation details in appendix.
>
> We thank the reviewer again for your very positive feedback! If you have any further questions or comments, please let us know, we are glad to respond.
>
> [1] Cross-Modal Relation-Aware Networks for Audio-Visual Event Localization

---

> > ### Comment · Reviewer_kKgn · 2023-08-20
> >
> > Thanks for the response. The line 144 says that $\Phi^a$ and $\Phi^b$ are for the *modal-agnostic features*, if they are different, could you explain why do they produce modal-agnostic features? I thought the modal-agnostic encoders would be a general feature extractor for any modalities.

---

> > > ### Author Response · Authors · 2023-08-20
> > >
> > > Thank you for your response. It is a very good question!
> > >
> > > As you mentioned, previous works often use a general feature encoder to represent different modalities and extract modal-agnostic features, and achieve good results in multi-modal implicit representation. However, in this work, although the **modal-agnostic encoders** used in our paper have different structures, our proposed **Cross-CPC** can offer cross-modal supervision, which means the fine-grained cross-modal predictions can force ${\Phi}^{a}$ and ${\Phi}^{b}$ to know which parts are more relevant to other modalities, and we believe these aspects of information can be regarded as **semantic information or modal-agnostic information**.
> > > Meanwhile, the **mutual information minimization loss** and **reconstruction loss** will make the modal-specific encoders (${\Psi}^{a}$ and ${\Psi}^{b}$) to extract **modal-specific information** that are unrelated to other modalities. By this way, our model can disentangle the original input information into two parts, one is modal-agnostic information, or semantic information (${\Phi}^{a}$ and ${\Phi}^{b}$), which we believe is highly related to other modalities. The other part is modal-specific features (${\Psi}^{a}$ and ${\Psi}^{b}$), which do not supply useful information for our unified representation, but is vital for original feature reconstruction.
> > >
> > > In conclusion, our proposed DICD module (especially the Cross-CPC module) will guide the ${\Phi}^{a}$ and ${\Phi}^{b}$ to extract modal-agnostic features from different modalities.

---

### Author Rebuttal · Authors · 2023-08-09

Dear reviewers,

We much appreciate for your acknowledgement of our work and helpful, insightful comments. Following the reviewers' suggestions, we have made a major revision of the paper and conducted a series of new experiments to address the reviewers' concerns. We have also updated two figures in the single-page PDF file as suggested by two reviewers. In the following, under each reviewer's comment, we address the concerns of the reviewers point by point.

---

### Decision · Program_Chairs · 2023-09-21

**Decision:**

Accept (poster)

**Comment:**

This paper introduces a novel idea supported by strong empirical results. The reviewers appreciate the valuable contribution of this study. In preparation for the camera-ready version, we kindly urge the authors to incorporate the feedback provided by the reviewers, which will undoubtedly further refine the quality and impact of the work.